# CTCF and Its Partners: Shaper of 3D Genome during Development

**DOI:** 10.3390/genes13081383

**Published:** 2022-08-02

**Authors:** Xiaoyue Sun, Jing Zhang, Chunwei Cao

**Affiliations:** 1Medical Research Center, Sun Yat-sen Memorial Hospital, Sun Yat-sen University, Guangzhou 510275, China; sunxy59@mail.sysu.edu.cn (X.S.); zhangj768@mail2.sysu.edu.cn (J.Z.); 2Guangdong Provincial Key Laboratory of Malignant Tumor Epigenetics and Gene Regulation, Sun Yat-sen Memorial Hospital, Sun Yat-sen University, Guangzhou 510275, China; 3Center for Reproductive Genetics and Reproductive Medicine, Sun Yat-sen Memorial Hospital, Sun Yat-sen University, Guangzhou 510275, China; 4Guangzhou Laboratory, Guangzhou 510320, China

**Keywords:** CTCF, 3D genome, development, protein partners, RNA partners, post-translational modifications

## Abstract

The 3D genome organization and its dynamic modulate genome function, playing a pivotal role in cell differentiation and development. CTCF and cohesin, acting as the core architectural components involved in chromatin looping and genome folding, can also recruit other protein or RNA partners to fine-tune genome structure during development. Moreover, systematic screening for partners of CTCF has been performed through high-throughput approaches. In particular, several novel protein and RNA partners, such as BHLHE40, WIZ, MAZ, Aire, MyoD, YY1, ZNF143, and Jpx, have been identified, and these partners are mostly implicated in transcriptional regulation and chromatin remodeling, offering a unique opportunity for dissecting their roles in higher-order chromatin organization by collaborating with CTCF and cohesin. Here, we review the latest advancements with an emphasis on features of CTCF partners and also discuss the specific functions of CTCF-associated complexes in chromatin structure modulation, which may extend our understanding of the functions of higher-order chromatin architecture in developmental processes.

## 1. Introduction

The genome stores the complete genetic information of living organisms, and the 3D genome is commonly regarded as genomic DNA sequence folding in three dimensions or higher-order chromatin organization inside a cell’s nucleus. Importantly, the dynamic of three-dimensional (3D) structure of the genome is closely associated with modulation of gene expression and genome function [1], which play critical roles in maintaining the normal developmental process [2]. Of note, the occurrence of many genetic diseases and even cancers, such as congenital limb malformations, autoimmune diseases, and breast cancer, have been found to be related to the variation of the 3D genome structure [3].

Hierarchically, the topological structures of 3D genome are organized at four levels, comprising chromosome territories, A/B compartments, topologically associating domains (TAD), and chromatin loops (e.g., enhancer-promoter interactions) (Figure 1A). Currently, there are two main techniques for investigating 3D genome folding, including imaging-based fluorescence in situ hybridization of DNA (DNA-FISH) and high-throughput chromosome conformation capture (Hi-C) [4,5]. DNA-FISH, which is the most commonly used technique for detection and validation of DNA sequence-specific contacts, can not only provide single-cell information but can visualize the spatial organization of chromosomes and genes in the nucleus. Unlike DNA-FISH, Hi-C is developed on the basis of high-throughput sequencing technology, which offers a high-resolution strategy for genome-wide DNA contact discovery. The computational analysis of Hi-C data can resolve the structure of the 3D genome in two spatial compartments between the entire genome, which are defined as compartment A and compartment B. The A compartment, associated with open chromatin and activation of gene expression, is usually located inside the nucleus. The B compartment, associated with closed chromatin and repression of gene expression, is often distributed at the periphery of the nucleus. Furthermore, a TAD refers to a specific genomic region, within which DNA sequences contact each other more frequently than with sequences outside the TAD. Chromatin loops, formed by long-range DNA sequence interactions, represent the basic structural unit of chromatin organization. Importantly, multiple variations of Hi-C and new techniques have been recently developed to extend the utility or resolution of the Hi-C method. Among them, non-enrichment methods, such as micrococcal nuclease chromosome conformation assay (Micro-C), split-pool recognition of interactions by tag extension (SPRITE), and genome architecture mapping (GAM), can also simultaneously capture chromatin conformation across the genome. Moreover, enrichment-based approaches, consisting of capture Hi-C (cHi-C), chromatin interaction analysis with paired-end tag (ChIA-PET), and DNA adenine methyltransferase identification (DamID), have been developed to investigate either specific regions of interest, or interactions mediated by specific protein [5].

CCCTC-binding factor (CTCF), serving as the central organizer, plays a key role in 3D genome organization. CTCF is highly conserved across vertebrate species but absent in plants, C. elegans, and yeast [6]. CTCF has a DNA-binding domain that binds to numerous target sites in the genome [7], and nearly 80% of CTCF binding sites share a specific 20 mer motif that is highly conserved in vertebrates [8]. Intriguingly, CTCF mainly binds at the TAD borders. In this regard, CTCF shapes 3D chromatin organization by determining the TAD boundaries [9]. Furthermore, the chromosome structure maintenance protein complex cohesin, another key player in regulating genome folding, has been shown to organize the 3D chromatin structure of mammalian genomes in cooperation with CTCF. Notably, the loop extrusion model, which depends on interaction and cooperation between cohesin and CTCF, has been built, and it clearly explains the formation of a subset of chromatin loops.

In addition to cohesin, extensive studies have revealed numerous partners, which work in close collaboration with CTCF in shaping the chromatin structures. Most of these partners are transcription factors that are involved in cell-type-specific transcriptional regulation and chromatin remodeling processes. CTCF is developmentally regulated, and interaction between the CTCF and these lineage-specific protein partners confers various functions during cell development and differentiation, ranging from embryonic stem cells (ESCs) to immune and muscle cells. Furthermore, recent studies emphasized that RNA can also interact with CTCF, and these interactions are essential for facilitating CTCF-mediated genome organization. The interplay between CTCF and its RNA partners has also been found to be involved in specific biological processes such as X chromosome inactivation. Here, we review recent advances in the discovery of CTCF partners and also discuss the specific functions of CTCF–partner complexes in the regulation of chromatin structure (Table 1), which should help to uncover epigenetic mechanisms of relevant developmental processes.

## 2. Cohesin: The Key Partner of CTCF

Regarding the formation of the basic three-dimensional structural loops, the “loop extrusion model” has been proposed by several groups (Figure 1B) [67,68,69]. According to the hypothesis, cohesin extrudes chromatin loops bidirectionally, and these chromatin loops will initially be small and increase over time until they are blocked by CTCF boundaries. Moreover, depending on the loop extrusion model, several modified models, including the walking model, pumping model, and scrunching model, have been proposed for the explanation of the loop extrusion process in eukaryotic mitosis cells, which improves our understanding of chromatin loop formation [70].

Cohesin, a circular protein complex, was originally identified in eukaryotic mitosis for its function in sister chromatids cohesion [71]. Moreover, cohesin also plays crucial roles in regulation of DNA loops formation and 3D genome dynamics. The cohesin complex typically contains four subunits: the ring-forming subunits SMC1, SMC3, and SCC1, and one HEAT repeat proteins associated with kleisins (HAWK) protein (STAG1 or STAG2) [72]. Cohesin binds genomic sequences in a cis manner and extrudes DNA bidirectionally to form chromatin loops until encountering the boundaries that are preferentially bound by CTCF [73]. Degradation of cohesin resulted in the elimination of nearly all loop domains, showing that cohesin is the key factor responsible for loop formation [74]. Indeed, several pieces of evidence have indicated that CTCF works together with cohesin in modulating chromatin structure. In particular, almost 90% of the cohesin ChIP-Seq peaks co-localized with CTCF binding sites [10,11] and ChIP-seq as well as Hi-C experiments found that cohesin and CTCF are enriched at the TAD boundary region, which support their coordinating function in TAD and loop establishment [12,13]. The orientations of CTCF motifs are divided into three categories: convergent orientations, same orientations, and divergent orientations. Most loop- or TAD-bound CTCF motif pairs appear in mutually convergent orientations and are critical for loop formation (Figure 1B). A single reversal of the orientation of the CTCF motif is sufficient to make the loop disappear and alter the DNA folding [75]. To visualize the dynamics of CTCF- and cohesin-mediated cycling, Hu et al. selected a relatively simple 505-kb TAD in mouse embryonic stem cells containing only one gene, *Fbn2*, as a model for study. Interestingly, direct observation of dynamic Fbn2 TAD chromatin loops via super-resolution live-cell imaging revealed that cohesin extrusion loops within TADs fail to bridge two CTCF boundaries approximately 92% of the time, suggesting that a single CTCF boundary can also create functional interactions [76].

Recently, several other factors, such as WAPL and NIPBL-MAU2, have been identified to participate in mediating loop formation or expansion. WAPL promotes the release of cohesin from DNA template, which restrains loop formation. Depletion of WAPL, in turn, increases the residence time of cohesin on chromatin, where cohesin bypasses CTCF binding sites and generates larger loops [14,15]. On the contrary, NIPBL forms a heterodimer with MAU2, and this complex works by loading cohesin onto DNA, which is required for loop extrusion. After removing NIPBL-MAU2, cohesin is barely detectable on chromatin in cells, showing that NIPBL-MAU2 functions as a key partner of cohesin. Moreover, NIPBL-MAU2 can stimulate the ATPase activity of cohesin, which is required for cohesin loading onto chromosomes [16].

## 3. Protein Partners of CTCF

### 3.1. Systemic Discovery of CTCF Partners

Recently, several studies have performed systematic investigation of CTCF’s partners. Hu et al. took advantage of 1306 ChIP-seq data for 431 human protein factors and characterized the genome-wide DNA-binding patterns of these factors, including transcription factors, histone variants, and histone-modifying enzymes, as well as CTCF, in 23 cell lines [52]. All factors with ChIP-seq data were screened by co-binding or co-localization analysis with human hyperconserved CTCF binding sites using computational methods. In addition to previously reported co-factors such as cohesin subunits (RAD21, SMC3), histone demethylase KDM5B, and transcription factors YY1 and ZNF143, a number of novel co-binding factors overlapping with CTCF-binding sites were identified. For example, RCOR1 and TEAD4 showed up to 40% overlapping with CTCF binding sites. These novel factors may be important candidate partners that are essential for the establishment of CTCF-mediated chromatin looping.

Furthermore, Marino et al. applied affinity purification coupled with high-resolution LC–MS/MS analysis for large-scale identification of CTCF-specific binding partners [60]. In the WiT49 cell line overexpressing CTCF, affinity purification using coimmunoprecipitation (co-IP) was first performed, and LC–MS/MS analysis of the pull-down products identified 90 high-confidence proteins putatively belonging to the CTCF interactome. These 90 proteins constitute a protein–protein interaction (PPI) network with specific functions enriched in chromatin binding, promoter-specific chromatin binding, and transcriptional regulation. Remarkably, proteins associated with ATP-dependent helicase activity have also been found, such as BRG1, which is the main ATPase subunit of the SWI/SNF chromatin remodeling complex. Likewise, Lehman et al. investigated CTCF’s interacting partners in MCF10A cells using LC–MS/MS, and their findings revealed that RNA-binding- and RNA-splicing-associated proteins (including snRNP, hnRNP, and serine-arginine rich proteins) were detected as the most prevalent CTCF binding partners, mainly localized in the interchromatin serine/arginine-rich splicing factor (SC-35) nuclear speckles [77].

### 3.2. Transcriptional Regulatory Protein Partners Related to Embryonic Stem Cell Development

CTCF has been reported to interact with RNA polymerase II both in vivo and in vitro [17], which is involved in the regulation of alternative splicing and transcription initiation processes [18]. In addition, a great number of recent studies indicated that CTCF also interacts with various transcription factors to regulate transcription and the three-dimensional genome structure, which is essential for embryonic stem (ES) cell self-renewal and differentiation. The pluripotency factor OCT4, acting as a key regulator at the top of the X chromosome inactivation (XCI) hierarchy, can interact with CTCF to regulate XCI by triggering X chromosome pairing and counting, which is indispensable to ES cell differentiation [19]. Moreover, CTCF, in addition to acting as an upstream transcription regulator of WD repeat domain 5 (Wdr5), can also physically bind to Wdr5, which plays important roles in maintaining ES cell pluripotency and somatic reprogramming [20]. WIZ, a zinc finger-containing protein, is also a structural regulator for the maintenance of stem cell properties and embryonic development [21,22]. Homozygous mutations in *WIZ* gene result in embryonic lethality, and heterozygous deletions increase anxiety-like behaviors in mice [22,23]. WIZ can form complexes with CTCF and cohesin at multiple sites in the genome, including enhancers, promoters, insulators, and anchors on DNA loops, which play major roles in negatively regulating cohesin occupancy on chromatin and DNA loop structures. Importantly, WIZ functions by direct interaction with CTCF rather than DNA or RNA dependence. However, WIZ knockout did not have a major effect on the A/B compartments switch and TAD reorganizing but increased the number of DNA loops and reduced its size, suggesting that WIZ should be a structural regulator of chromatin loops [21]. In addition, TAF3, a TBP-associated core promoter factor, is required for ES cell differentiation towards the endoderm lineage, preventing the premature formation of neuroectoderm and mesoderm. Liu et al. demonstrated that CTCF directly recruited TAF3 to promoter-distal sites in ES cells to mediate long-range chromatin contacts, thereby supporting a finely balanced transcriptional program of pluripotency (Figure 2) [24].

### 3.3. Transcriptional Regulatory Protein Partners Regulating Immune Cell Development

T and B cells comprise the main forces of adaptive immunity of the immune system, and T cells play a major role in the body’s anti-infection, anti-tumor, and autoimmune diseases. Naïve T cells can differentiate into Th1, Th2, and Th17 cells expressing IFN-γ, IL-4, and IL-17, respectively. Kim et al. showed that Th17 lineage differentiation is restrained by the Th2 locus, and the regulation is attributed to the interaction between Oct-1 and CTCF, which mediates the interchromosomal contacts between the locus control region (LCR) of the Th2 cytokine locus on chromosome 11 and the IL-17 locus on chromosome 1 [25]. The high mobility group (HMG) transcription factor TCF-1 functions at the early thymic progenitor (ETP) stage by regulating Gata3 and Bcl11b expression and remains highly expressed until maturation, which is essential for early T-cell development [26]. Wang et al. demonstrated that TCF-1 and CTCF co-occupy recombined TAD boundaries during T-cell development, weakening the insulation between adjacent neighbors and enhancing the interaction between regulatory elements and target genes located on previously insulating domains. The promotion of chromatin interactions mediated by TCF-1 is associated with the deposition of the active enhancer marker H3K27ac and the recruitment of NIPBL [27]. What is more, Bansal et al. indicated that in medullary thymic epithelial cells (mTECs), the transcription factor Aire controls immune tolerance by driving abundant gene expression and has broad effects on the organization of 3D chromatin structure [28]. Aire preferentially localizes to superenhancers and promotes the formation of superenhancer-promoter loops to regulate chromatin structure. On the one hand, Aire supports the chromatin transcribability by promoting the accumulation of cohesin and mediators in superenhancers to facilitate the transition from the inactive B chromatin state to the active A state; on the other hand, it counteracts structural loops by expelling CTCFs at TAD or CD boundaries [28]. Zinc finger protein ZNF143 is a sequence-specific transcriptional activator of Pol II and Pol III and is associated with T lymphocytic leukemia (TLL) [29,30]. In addition, ZNF143 is also identified as a novel chromatin loop regulator, which has the capacities to link promoters to anchors for chromatin interactions of distal regulatory elements [31]. Zhou et al. demonstrated that ZNF143 is a key regulator of CTCF-binding promoter–enhancer loops that are essential for maintaining the mouse hematopoietic stem and progenitor cell integrity. Interestingly, the specific spacing between most ZNF143 and CTCF binding sites is 37 bp in the murine genome, implying that ZNF143 may act as a regulator of CTCF by mediating its DNA binding ability [32]. Moreover, Lee et al. reported that interaction of the transcriptional cofactor LDB1 complex with CTCF underlies erythroid lineage-specific long-range enhancer interactions. Significantly, LDB1 and CTCF mediate the activation of *Car2* in erythrocytes by binding to the enhancer and promoter upstream of the *Car2* gene, respectively [33].

### 3.4. Transcriptional Regulatory Protein Partners Associated with Muscle Cell Development

MyoD and Myf5, acting as fundamental helix-loop-helix transcription factors, are required for myogenic initiation during early embryogenesis and functionally complement each other [34]. MyoD inactivation in mice had no apparent effect on muscle development [35] but resulted in severely deficient muscle regeneration [36] and reduced differentiation potential [37]. Wang et al. have shown that MyoD cooperated with CTCF, thus driving the formation of different types of chromatin loops in muscle cells. Notably, four types of chromatin loops, namely, MyoD-MyoD (noCTCF), MyoD-MyoD (CTCF), MyoD-CTCF, and CTCF-CTCF, have been resolved. The MyoD-binding loop was significantly shorter than the CTCF-CTCF loop, with the MyoD-MyoD (noCTCF) loop being the shortest, and MyoD inactivation significantly reduced loop strength in all four types of chromatin loops. Taken together, MyoD functionally contributes to the formation of MyoD-bound and CTCF-bound chromatin loops. Of note, MyoD regulates chromatin loops independently of H3K27ac levels, suggesting that MyoD should be a key organizer in establishing the unique 3D genomic architecture of muscle cells [38].

### 3.5. Transcriptional Regulatory Protein Partners Involved in Multiple Developmental Processes

The zinc finger transcription factor Yin Yang 1 (YY1) is essential for both early embryogenesis and adult tissue development [39], and biallelic loss of function variants in YY1 cause embryonic lethality in mice [40]. In embryonic stem cells, YY1 activates transcription by targeting promoters and super-enhancers through the BAF complex [41]. YY1 is also a master regulator to coordinate multidimensional epigenetic crosstalk associated with expanded pluripotency, and depletion of YY1 disrupts specific enhancer–promoter interactions in expanded pluripotent stem cells (EPSC) [42]. Further, YY1 directly interacts with CTCF and they work together to regulate X chromosome binary switch [43]. Moreover, YY1 deletion in EPSCs reduces DNA methylation, promotes CTCF binding to hypomethylated DNA regions, and promotes gene expression. In B cells, YY1 mediated long-range DNA contacts [44] and is necessary for the formation of specific 3D interactions [45]. Furthermore, YY1 is also a key regulator of neuron differentiation of neural progenitor cells (NPC) to myelinated oligodendrocytes [46]. YY1 can function as a structural protein, linking NPC-specific genes and enhancers, and is implicated in regulation of 3D interactions. More importantly, as a CTCF’s partner, the interactions mediated by YY1 between regulatory elements are often in CTCF-anchored constituent loops [47].

The Y-box DNA/RNA binding factor (YB-1) is a multifunctional protein involved in transcription, replication, and RNA processing, and several studies have identified it as a CTCF partner. Chernukhin et al. demonstrated that YB-1 can physically and functionally interact with CTCF to repress the transcription of c-myc [48]. In addition, Klenova et al. found that YB-1 cooperated with CTCF to modulate the 5-HTT polymorphic intron 2 enhancer associated with nervous system development [49]. In addition, Wang et al. demonstrated that CTCF is essential for peripheral nerve remyelination and Schwann cell myelination. On one hand, CTCF establishes chromatin loops that promote the expression of the key pro-myelinating factor EGR2. On the other hand, CTCF interacts with SUZ12, a component of polycomb-repressive-complex 2 (PRC2), to suppress the differentiation inhibitory pathway of Schwann cells [50].

### 3.6. Transcriptional Regulatory Protein Partners Showing Potential Roles in 3D Genome Organization and Transcriptional Regulation

Xiao et al. have shown that Myc-associated zinc finger protein (MAZ) is a physically interacting partner of CTCF, and it can function as a genome architecture protein, one that participates in genome organization. Remarkably, MAZ shares core properties with CTCF, including insulation activity and interaction with cohesin subunit Rad21, supporting the fact that MAZ and CTCF have complementary roles in organizing genome structure [51]. In addition, ChIP-seq and co-IP experiments showed that most of the basic helix-loop-helix family member e40 (BHLHE40) binding sites are also occupied by CTCF, and BHLHE40 can physically interact with CTCF, indicating that BHLHE40 should be a CTCF partner [52]. Furthermore, loss of BHLHE40 results in a reduced number of CTCF binding sites, decreased CTCF loop strength, and disruption of CTCF-mediated long-range chromatin interactions, demonstrating that BHLHE40 acts as a partner to regulate CTCF-mediated chromatin interactome.

In addition, several other protein partners of CTCF have been identified; nevertheless, the detailed mechanisms underlying the roles of CTCF/partner complex in regulation of 3D genome structure still require further information. For example, Kaiso is a bimodal transcription factor that recognizes methylated CpG dinucleotides or conserved unmethylated sequences (TNGCAGGA, Kaiso binding site). The Kaiso binding sites exist adjacent to CTCF binding sites, and the Kaiso–CTCF interaction negatively regulates CTCF insulator activity [53]. Additionally, CTCF directly interacts with transcription factor RFX and transcription coactivator CIITA to form a trimeric complex to regulate HLA-DRB1 and HLA-DQA1 gene transcription [54,55].

### 3.7. Chromatin Remodeling Associated Protein Partners

The nucleosome is the basic unit of chromatin, consisting of histone octamers (two copies of H2A, H2B, H3, and H4) and 146 base pairs (bp) of DNA. In addition to these classic histone variants, many other variants have been found, such as H2A.X, H2A.Z, H3.5, and H4.G [78]. Moreover, a great number of proteins, which have the capacities of modifying the chromatin architecture, have been identified, and these proteins are defined as chromatin-remodeling-associated proteins [79,80]. Until now, studies have identified several CTCF’s partners that are related to chromatin remodeling. H2A.Z is a histone variant that shows specific properties in regulating higher-order chromatin structures [81]. In vitro studies have shown that the structure of nucleosomes is not static but undergoes spontaneous structural transitions, including DNA respiration (spontaneous opening of DNA ends on nucleosomes) and open states (opening of interfaces between histone subcomplexes), and this dynamic change of nucleosomes is called nucleosome unwrapping. By employing micrococcal nuclease (MNase) digestion of crosslinked chromatin, chromatin immunoprecipitation, and the paired-end sequencing (MNase-X-ChIP-seq) approach, Wen et al. investigated the genome-wide unwrapping state of H2A.Z nucleosomes in mouse embryonic stem cells. Interestingly, compared with canonical nucleosomes, H2A.Z is enriched with nucleosome unwrapping, indicating that H2A.Z is essential to nucleosome unfolding, and H2A.Z may affect CTCF binding regulation and gene expression by modulating the unwrapping states of nucleosomes [56]. What is more, Ishihara et al. demonstrated that the SNF2-like chromatin domain helicase protein (CHD8), containing two chromatin organization modifier domains, interacts with CTCF and then regulates the insulator activity of CTCF, which is associated with epigenetic regulation and chromatin organization remodeling [57]. Likewise, Qiu et al. reported that bromodomain PHD finger transcription factor (BPTF), a member of the nucleosome–remodeling factor (NURF) complex, was also found to interact with CTCF [58]. NURF, regarded as a nucleosome remodeling enzyme, is involved in regulating the higher-order structure of chromatin [59]. Therefore, the interplay between CTCF and BPTF may play specific roles in 3D genome shaping. Moreover, Marino et al. reported that BRG1, the major ATPase subunit of the chromatin remodeling complex SWI/SNF, interacts with CTCF [60]. BRG1 is enriched at TAD boundaries, and its inactivation significantly reduces TAD boundary strength and alters the long-range genomic interactions [61], suggesting that disrupted interaction between CTCF and BRG1 impairs chromatin organization at the TAD boundary. Furthermore, Lutz et al. reported that CTCF retained histone deacetylase activity by directly interacting with a chromatin remodeling factor, such as SIN3A [62].

### 3.8. Interplay between Nuclear Receptor and CTCF

The nuclear receptor superfamily is a family of ligand-activated transcription factors that regulate cell growth and differentiation by establishing links between signaling molecules and transcriptional responses [82]. Warwick et al. demonstrated that activation of the vitamin D receptor (VDR) and its high-affinity ligand 1,25(OH)2D3 induced cellular 3D chromatin changes. Interestingly, their results indicated that more than 3000 CTCF interactions were altered, and VDR binding sites and vitamin D target genes are preferentially located at loop anchors, implying the potential interaction between CTCF and VDR [63]. Furthermore, upon estrogen stimulation in breast cancer cells, CTCF binds to enhancer regions and prevents the formation of estrogen receptor (ER)-mediated chromatin loops to regulate ER target transcription [64]. During chromatin remodeling, switching between active A and inactive B compartments in endocrine-resistant breast cancer is associated with reduced ER binding and aberrant ER-mediated enhancer–promoter interactions [65]. Therefore, VDR and ER may be partners of CTCF, and their interactions may play important roles in regulating the organization of the 3D genome.

## 4. RNA Partners of CTCF

In addition to binding to DNA, several recent studies confirmed that CTCF also has the ability to bind to RNA. Moreover, CTCF targets thousands of transcripts throughout the genome and has a higher binding affinity for RNA than DNA [83]. ZF1 and ZF10 domains, but not the major DNA binding domains (ZF3-7 domains), are responsible for the RNA-binding property of CTCF [84]. Hansen et al. reports that CTCF’s RNA-binding region (RBR) plays crucial roles in CTCF clustering in vivo and is associated with chromatin loop formation. More importantly, RBR suppression in mESCs causes disruption in about half of the chromatin loops, called RBRi-dependent loops [85].

Currently, high-throughput-sequencing-based methods, such as UV-crosslinking immunoprecipitation and deep sequencing (CLIP-seq), have been developed to identify CTCF-bound transcripts systemically [83]. Additionally, Kuang et al. discovered a novel RNA-binding motif (AGAUNGGA) of CTCF and identified 4925 candidate CTCF-binding lncRNAs by a deep learning model DeepLncCTCF, extending our understandings of CTCF in 3D genome organization [86]. Furthermore, to measure higher-order RNA and DNA contacts within 3D structures, the RNA and DNA split-pool via label-extended recognition interaction (RD-SPRITE) method was developed. Depending on this method, hundreds of noncoding RNAs (ncRNAs) were found to form regions of high concentration within the nucleus, and these higher-order RNA-chromatin structures are related to regulation of long-range DNA contacts, heterochromatin assembly, and gene expression [87].

Until now, several CTCF’s RNA partners have been identified, and their regulatory roles in chromatin structure provide novel insights into 3D genome organization. X-chromosome inactivation (XCI) is a classical epigenetic reprogramming process that is essential for mammalian development [88]. Studies indicated that CTCF is implicated in the XCI process, which is largely driven by Tsix, Xite, and Xist RNAs, and CTCF directly interacts with these RNAs in the X inactivation center during XCI, thereby mediating long-range chromosomal interactions [83,89,90]. Jpx RNA, another CTCF partner, also plays roles in regulating the initiation of X chromosome inactivation (XCI) by expelling CTCF from the Xist promoter [89]. Notably, a recent study demonstrated that Jpx/CTCF complex modulates the chromatin structure on a genome-wide manner, not limited to XCI. Importantly, Jpx can act as a CTCF release factor and determine the anchoring selectivity of CTCF. Specifically, Jpx selectively binds to low-affinity CTCF motifs and expels CTCF through competitive inhibition (Figure 3). Thus, knockdown of Jpx RNA results in substantial changes in chromosomal loops, most likely due to the ectopic CTCF binding [90].

In addition, interactions between CTCF and multiple ncRNAs, such as the steroid receptor RNA activator (SRA), Wrap53, HOTTIP, and GATA6-AS1, have been identified, playing potential roles in shaping 3D genome structure. For example, Yao et al. revealed that CTCF/SRA/p68 (DEAD-box RNA helicase) complex can stabilize the interaction between CTCF and cohesin [91]. CTCF regulates p53 expression by physically interacting with Wrap53 RNA, which is the natural antisense transcript of p53. Deletion of CTCF not only resulted in a decrease in p53 mRNA levels, but also in Wrap53 levels [92]. The HOXA transcript at the distal tip (HOTTIP) was previously identified as a lncRNA located at the 5′ end of the HOXA locus [93]. Luo et al. found that overexpression of HOTTIP restores CTCF-mediated HOXA TAD and causes leukemogenesis, but the mechanism by which HOTTIP regulates the CTCF boundary activity is unclear [94]. In addition, the lncRNA GATA6-AS1 triplex-forming sites were recently found to be enriched at the TAD boundary during cardiac differentiation, implying the interaction between GATA6-AS1 and CTCF [95]. Furthermore, Miyata et al. reported that ncRNAs can change the genomic distribution of CTCF, interfering with the expression of inflammatory genes in aging and cancer [96].

## 5. Post-Translational Modifications of CTCF

Protein post-translational modifications (PTMs), functioning as a key mechanism for increasing proteome diversity, play critical roles in nearly all biological processes. Indeed, many studies indicate that PTM enzymes, such as PARP, SUMO, CK2, PLK1, and LATS, are also CTCF’s protein partners. In fact, CTCF is regulated by serving as a substrate for these PTM enzymes. PTMs of CTCF are associated with dynamic regulation of the stability and function of CTCF in response to an external or internal stimulus (Figure 4A,B).

Poly(ADP-ribosyl) ation (PARylation) is mediated by poly(ADP-ribose) polymerase (PARP) [97], and an increasing number of studies suggest that PARylation is involved in CTCF function regulation. The N-terminal domain of CTCF is a preferred target for PARylation in vitro [98], and CTCF/PARP1 complex can function through a dynamic reversible PARylation modification model to regulate CTCF function [99], and is associated with the contacts between clock-controlled genes and lamina-associated chromatin [100]. Intriguingly, the PARylated CTCF isoform (180 kDa) preferentially located at nucleolus and PARylation of CTCF represses nucleolar transcription [101]. Furthermore, CTCF PARylation affects the binding of CTCF to chromatin, and the decrease in CTCF PARylation has been reported to be linked to breast tumorigenesis and cell proliferation [102,103].

CTCF protein can be modified by small ubiquitin-like proteins, including SUMO 1, 2, and 3. Currently, two major SUMOylation sites of CTCF, located in the COOH and NH2 terminal domains, separately, have been resolved. SUMOylation of CTCF commonly contributes to its repressive functions, such as repression of the cMYC P2 promoter activity [104]. Specially, a 107-amino-acid domain was identified in the N-terminal region of CTCF that activates transcription and depolymerizes chromatin, and complete sumoylation of this domain abolishes the transcriptional activity of CTCF and prevents chromatin opening [105]. Furthermore, stress-induced hypoxic desumoylation of lysines 74 and 689 in CTCF proteins regulated both the activity of CTCF and its downstream target genes [106].

Moreover, it is widely accepted that phosphorylation is a functional determinant of transcription factors and appears to be one of the most studied forms of PTM. The phosphorylation profile of CTCF is dynamic during development and cell differentiation [107]. CK2-mediated phosphorylation at several functional phosphorylation sites within CTCF C-terminal region can convert CTCF inhibitory function to activating function [108]. In addition, amino acid residues (Thr289, Thr317, Thr346, Thr374, Ser402, Ser461, and Thr518) of the CTCF linker domain are phosphorylated during mitosis to regulate its DNA-binding activity [109]. In addition, another Polo-like kinase 1 (PLK1)-mediated phosphorylation of CTCF at serine 224 (Ser224-P) is enriched in the G2/M phase of the cell cycle, especially at pericentric regions. Of note, the CTCF phospho-depletion mutant S224E resulted in dysregulation of hundreds of target genes, including p53 and p21 [110]. CTCF was found to be a substrate for LATS kinase, and cellular-stress-induced activity of LATS directly phosphorylates CTCF’s zinc finger (ZF) linker and selectively dissociates CTCF from a small fraction of its genomic binding sites, impairing its DNA-binding activity [111]. Therefore, external signals may modulate 3D genome structure through the phosphorylation of CTCF’s ZF linker.

## 6. Conclusions and Perspectives

The orderly folding of the genome in the three-dimensional space of the nucleus is critical for developmental processes, and the 3D genome folding is dynamic at different developmental stages. Significantly, the loop extrusion model explains the basic features of loop formation and genome folding, opening a new window into chromatin organization and genome architecture. According to this model, two key players CTCF and cohesin work together to regulate genome folding into TADs and loops. Indeed, a large number of studies demonstrated that protein factors and RNAs are implicated in chromatin structure regulation by interacting with CTCF. These partners act to shape nuclear structure in a general or a specific manner, and these new findings provide crucial information for decoding the 3D genome structure and its dynamics, as well as elucidating a causal relationship between dysregulated high-order chromatin structure and developmental processes.

At present, a large number of protein and RNA partners of CTCF have been identified through systemic screening approaches, including ChIP-seq, LC–MS techniques, and the DeepLncCTCF model [52,60,77,86]. Nonetheless, how these partners positively or negatively regulate CTCF-mediated functions and the specific mechanisms underlying their regulatory roles in 3D genome organization remain to be uncovered in the future. In addition, partners, who function in a cell/tissue specific or a developmental stage specific manner, have attracted more attention in recent years. MyoD and Aire, for instance, mainly function in muscle and immune cells, respectively. These studies, revealing the partners’ specific roles in CTCF function regulation, extend our knowledge about the roles of 3D genome organization in development, cell differentiation, and pathogenesis. Moreover, these findings help to address the question of how variability of the high-order chromatin is structured among different cell types.

Compared to protein partners, findings related to RNA partners of CTCF are relatively limited. However, a growing amount of evidence indicates that ncRNAs also play crucial roles in shaping the 3D genome. Early in 1989, Nickerson et al. made the concept that RNA may function as a structural component, and participate in organizing the higher-order structure of chromatin [112]. Recently, Saldaña-Meyer et al. performed a systemic investigation of function of a CTCF–RNA complex in 3D genome organization, and they suggested that most RNA molecules may show structural and stabilizing roles, and also have the abilities to modulate protein–protein interactions [84]. Specially, Bouwman et al. reviews and highlights the relationship between nuclear RNAs (nucRNAs) and 3D genome shaping. NucRNAs play roles in the 3D genome shaping in various manners, either acting locally on a specific region, or globally on the genome. Intriguingly, recent studies suggest that the formation of global and local nucRNAs gradients might be responsible for the 3D genome shaping [113]. However, more evidence is required to confirm the hypothesis and investigate the potential roles of CTCF–nucRNA interaction in higher-order chromatin structure modulation.

Liquid–liquid phase separation (LLPS) is a process that separates a homogeneous liquid solution of macromolecules, such as proteins or nucleic acids, into two distinct phases: dense and dilute [114]. Numerous studies show that many important biological processes, such as DNA replication, DNA damage repair, transcription, and RNA processing, occur in biomolecular condensates formed by LLPS. Through LLPS, transcription machinery assembles into transcriptional condensates at super-enhancers and drives the expression of downstream target genes [115,116]. Recent studies have found that CTCF-mediated chromatin looping may be a key prerequisite for the assembly of phase-separated transcriptional condensates [117], and its protein partner harboring LLPS property might modulate phase separation of CTCF. Wei et al. showed that CTCF can mediate long-distance interactions between A compartment through RYBP-dependent phase separation. Furthermore, Wei et al. demonstrated that CTCF-mediated phase separation can regulate the pluripotency of embryonic stem cells [66]. In addition, Wang et al. found that Oct4 can regulate TAD recombination and promote somatic reprogramming via a phase-separation mechanism [118]. Given that Oct4 was previously found to interact with CTCF [19], Oct4 may also be a CTCF’s partner to regulate the 3D genome in ES cells through the phase separation process. Therefore, we point out that discovery of partners that can promote LLPS of CTCF would be a new research direction.

Taken together, the function of the CTCF–protein and CTCF–RNA interactions in the genome organization has become an important research focus, and recent findings advance our understanding of CTCF/partner function in shaping 3D genome structure. Furthermore, experimental and computational techniques are improving rapidly. Thus, more partners, functioning as structural components of the nucleus, are expected to be explored.

## Figures and Tables

**Figure 1 genes-13-01383-f001:**
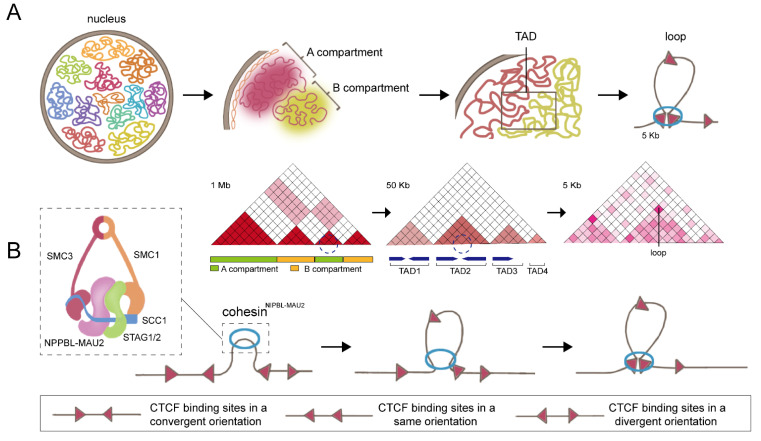
Three-dimensional genome organization and loop extrusion model. (**A**) The topological structures of the 3D genome are organized at four levels, comprising chromosome territories, A/B compartments, TAD, and chromatin loops. (**B**) DNA loop extrusion model. Cohesin binds to DNA and begins to extrude symmetrically until it encounters a convergent-oriented CTCF to form a chromatin loop.

**Figure 2 genes-13-01383-f002:**
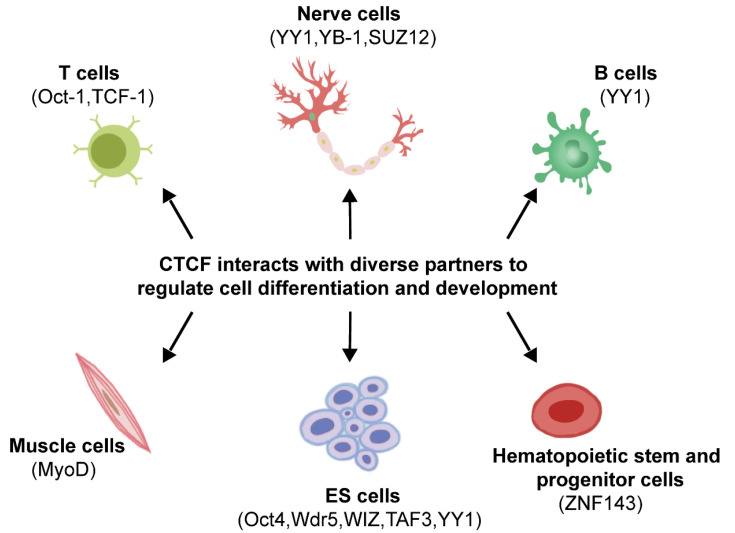
CTCF interacts with its protein partners to regulate cell differentiation and development.

**Figure 3 genes-13-01383-f003:**
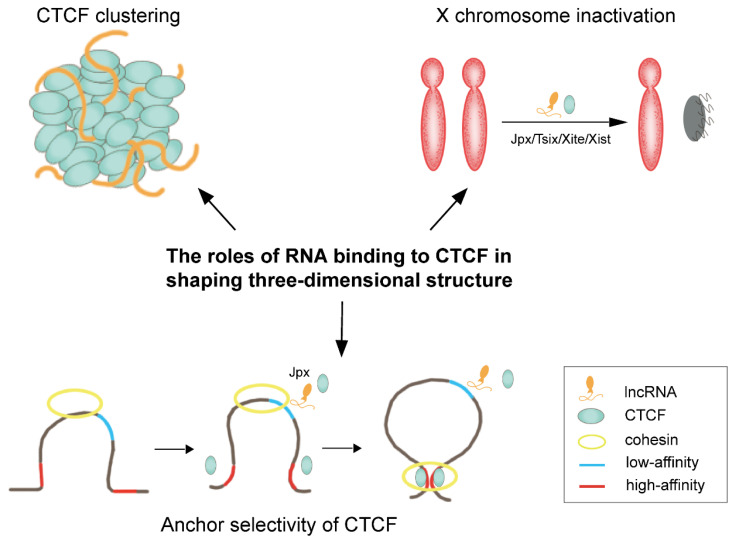
RNA/CTCF complexes play important roles in organizing chromatin structure.

**Figure 4 genes-13-01383-f004:**
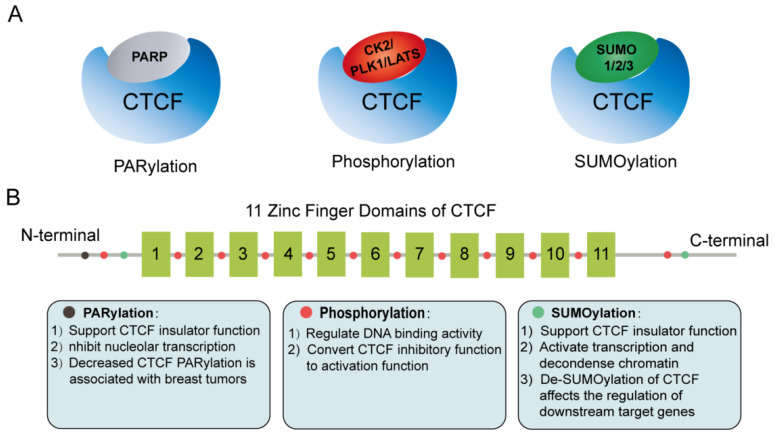
Post-translational modification (PTM) enzymes are CTCF’s partners. (**A**) CTCF is regulated by three types of PTM, namely, PARylation, SUMOylation, and phosphorylation. (**B**) PTM sites in CTCF protein, and their roles in modulating CTCF insulator function.

**Table 1 genes-13-01383-t001:** The protein partners of CTCF in the 3D genome.

Functional Classification	Partners	Cells	Experimental Evidence	Functional Description	Ref.
Loop extrusion	Cohesion	ES cells, Jurkat cells	Co-IP, Chip	binding to DNA and extruding loops	[10,11,12,13]
	WAPL	-	-	releasing cohesin from DNA templates	[14,15]
	NIPBL	-	-	stimulating the ATPase activity of cohesin	[16]
Transcription	RNA polymerase II	HeLa cells, K562 cells	Co-IP, Chip	regulating transcription and alternative splicing	[17,18]
ES cell development	Oct4	ES cells	Co-IP, Chip	regulating XCI by triggering X chromosome pairing and counting	[19]
	Wdr5	ES cells	Co-IP, Chip	acting as a downstream target of CTCF, and maintaining ES cell pluripotency and somatic reprogramming	[20]
	WIZ	ES cells	Co-IP, Chip	acting as a structural regulator of DNA loops, and maintaining ES cell pluripotency and embryonic development	[21,22,23]
	TAF3	ES cells	Co-IP, Chip	mediating long-range chromatin regulation, supporting ES cells differentiate into endoderm	[24]
Immune cell development	Oct-1	Naive T cells	Co-IP, Chip	regulating naive T-cell differentiation to the Th17 lineage by mediating the contacts of the Th2 locus with the IL-17 locus	[25]
	TCF-1	T cells	Chip	regulating early T-cell development by modulating the TAD boundary formation and long-range chromatin interactions	[26,27]
	Arie	Medullary thymic epithelial cells	Co-IP, Chip	controlling immunological tolerance by promoting superenhancer–promoter loop formation	[28]
	ZNF143	Hematopoietic stem and progenitor cells	Co-IP, Chip	maintaining the integrity of mouse hematopoietic stem and progenitor cells by regulating CTCF-bound promoter–enhancer loops	[29,30,31,32]
	LDB1	MEL cells	Co-IP, Chip	mediating erythroid lineage-specific long-range enhancer interactions	[33]
Muscle cell development	MyoD	Muscle cells	Chip	forming distinct chromatin loops with CTCF, and building the unique 3D genome structure of muscle cells	[34,35,36,37,38]
Multiple developmental processes	YY1	ES cells, B cells, NPC cells	Co-IP, Chip	acting as a structural protein of the 3D genome, and mediating long-range DNA contacts	[39,40,41,42,43,44,45,46,47]
	YB-1	Hela cells,	Co-IP, Chip	inhibiting c-myc transcription, and regulating the 5-HTT polymorphic intron 2 enhancer	[48,49]
	SUZ12	Schwann cells	Co-IP, Chip	suppressing the differentiation inhibitory-pathway in Schwann cells	[50]
Potential roles in development	MAZ	K562, HepG2, HeLa	Co-IP, Chip	acting as a structural proteins of the 3D genome, and stabilizing CTCF binding to DNA	[51]
	BHLHE40	HeLa cells	Co-IP, Chip	regulating CTCF genome-wide distribution and long-range chromatin interactions	[52]
	Kaiso	HeLa cells,	Co-IP, Chip	regulating CTCF insulator activity	[53]
	RFX	Raji cells	Co-IP	regulating HLA-DRB1 and HLA-DQA1 gene transcription	[54,55]
	CIITA	Raji cells	Co-IP	regulating HLA-DRB1 and HLA-DQA1 gene transcription	[54,55]
Chromatin remodeling process	H2A.Z	ES cells	Chip	modulating nucleosome unwrapping and CTCF binding sites	[56]
	CDH8	HeLa cells	GST, Chip	regulating CTCF insulator function	[57]
	BPTF	ES cells	Co-IP, Chip	participating in chromatin remodeling, and regulating Klf4 binding near CTCF sites	[58,59]
	BRG1	WiT49, HeLa	Co-IP, Chip	mediating long-range chromatin interactions	[60,61]
	SIN3A	HeLa cells	GST, Chip	modulating the histone deacetylase activity of CTCF	[62]
Nuclear receptor	VDR	THP-1	Chip	inducing 3D chromatin changes upon activation by 1,25(OH)2D3	[63]
	ER	MCF-7	Chip	inducing 3D chromatin changes upon estrogen activation	[64,65]
Liquid–liquid phase separation	RYBP	ES cells	Co-IP, Chip	mediating long-distance interactions between A compartment by phase separation, and regulating the pluripotency of ES cells	[66]

Abbreviations: Co-IP, co-immunoprecipitation; GST, glutathione-S-transferase pull down assay; Chip, chromatin immunoprecipitation; CHART-seq, RNA target sequencing.

## Data Availability

Not applicable.

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
