# Peer review of "CTCF and Its Partners: Shaper of 3D Genome during Development"

_genes, 2022, doi:10.3390/genes13081383_

Round 1

Reviewer 1 Report

This review describes the recent advancements with an emphasis on features of CTCF partners and will also discuss the specific functions of CTCF-associated complexes in chromatin structure modulation, which should help to uncover epigenetic mechanisms of relevant developmental processes. The manuscript is well structured and well discussed. However, some points should be checked and corrected before its acceptance in this journal. 

Therefore, according to my comments, I recommended the publication of the paper after minor revision.

·   The study's background should be clearly stated. Describe the introduction and review of the work (Please add more information).

·         The MS English needs to be improved. The article's English must be carefully checked for grammatical errors.

Author Response

Dear Reviewer:

Thank you for your comments concerning our manuscript“CTCF and its partners: shaper of 3D genome during development”(genes-1834004). These comments are all valuable and very helpful for revising and improving our paper, as well as the important guiding significance to our researches. We have studied comments carefully and have made correction which we hope meet with approval.

All modified parts are marked in red in the paper. You can find my itemized responses below and my revisions/corrections in the resubmission.

We deeply appreciate your consideration of our manuscript. Thank you and best regards.

Reviewer 2 Report

In the manuscript "CTCF and its partners: shaper of 3D genome during development", written by Sun X, Zhang J and Cao C, the role of CTCF, acting in modelling the 3D structure of the chromatin, is presented. The manuscript in the first part describes the basic facts about 3D structure of the genome and new techniques used for detection of the molecules involved in their structure. In the following paragraphs the main partners of CTCF are presented: cohesins, different transcriptional factors which also have a role in loop formation and are important in different developmental processes and RNA partners. At the end, posttranslational modifications of CTCF are described. Conclusions are given together with perspectives, directions of investigations in the future.

The manuscript is well organized, systematic, presents a huge amount of data which are up to date.

Minor comments:

line 50: sentences reorganization: ...usually located inside the nucleus. Whereas....

line 74: sentence reorganization: invariant across...

line 77: sentence reorganization

line 97: in eukaryotic mitosis cells

line 102: extrudes – needs object

line 109: which support

line112: sentence reorganization

lines 113, 201, 252, 330: sentence should not begin with And

line 115: Fbn2 explanation

line 117: omit coma

line 132: sentence reorganization

line 134: sentence reorganization

line 152: MCF10A are not "normal" cells

line 160,293: a great number

line 163, 232, 296 375: after among object is needed

line 169: can WIZ be defined as a key regulator?

line 172: sites in the genome

line 176: sentence reorganization

line 178: sentence reorganization 3 times DNA loops in the same sentence

line 182: recruited

line 189: sentence reorganization

line 191: showed

line 199: what

line 218: sentence reorganization

line 234: loop intensity?

line 240: both,

line 263: on one hand,

line 274: explanation of what is BHLHE40

line 300: explain the unfolding state of nucleosomes

line 306: regulated

line 338: for DNA. Not through... sentence reorganization

line 343: which are

line 345: have been developed,  sentence reorganization

line 348: interacted with

line 356: until

line 358: classical

line 374: sentence reorganization

line 376: stabilize

line 393: sentence reorganization

line 404: explanation of interlaminar activity

line 406: nucleolus

line 410: omit also

line 413: cMYC

line 414: specially

line 415: ...can activate transcription... better explanation

 line 416: desumoylation

line 430: a substrate for LATS kinase

line 442: sentence reorganization, no need for questions , a great number

line 458: sentence reorganization

line 462: Nickerson... made the concept

line 470: on the genome

line 484: question

All references should have all the authors (some are at al.).

Author Response

(The authors gave the same response as above.)
